# Taorong-type Baijiu starter: Analysis of fungal community and metabolic characteristics of middle-temperature Daqu and high-temperature Daqu

Yanbo Liu[1,2,3,4,5], Xin Li[1,4,5], Haideng Li[1,6], Huimin Zhang[1,4,5], Xiangkun Shen[7], Lixin Zhang[3], Suna Han[2], Chunmei Pan[1,4,5]*

1 College of Food and Biological Engineering(Liquor College), Henan University of Animal Husbandry and Economy, Zhengzhou, China, 2 Postdoctoral Programme, Henan Yangshao Distillery Co., Ltd., Mianchi, China, 3 School of Life Sciences, Henan University, Kaifeng, China, 4 Henan Liquor Style Engineering Technology Research Center, Henan University of Animal Husbandry and Economy, Zhengzhou, China, 5 Zhengzhou Key Laboratory of Liquor Brewing Microbial Technology, Henan University of Animal Husbandry and Economy, Zhengzhou, China, 6 College of Biological Engineering, Henan University of Technology, Zhengzhou, China, 7 Henan Food Industry Science Research Institute Co., Ltd., Zhengzhou, China

* sige518888@163.com

**Data Availability Statement:** Relevant raw data has been uploaded to NCBI database and the accession numbers is PRJNA861706.

## Abstract

To study the difference between the fungal community compositional and fragrance components in medium- and high-Temperature Taorong-type Baijiu Daqu. The microbial communities and fragrance components of Taorong-type Baijiu Daqu were analyzed using high-throughput sequencing (HTS) and headspace-solid-phase microextraction-gas chromatography-mass spectrometry (HS–SPME–GC–MS). With an abundance at the phylum and genus levels $\geq 0.01\%$ as the threshold, 3 phyla, Mucoromycota, Ascomycota, and Basidiomycota, were found in both medium- and high-temperature Daqu, but their abundances differed. At the genus level, 15 and 13 genera were recognized. *Rhizopus* (72.40%) and *Thermomyces* (53.32%) accounted for the most significant proportions in medium-temperature and high-temperature Daqu, respectively. Medium-temperature Daqu and high-temperature Daqu were found to have 40 and 29 fragrance components, respectively and contained the highest proportions of pyrazines (53.12%) and acids (32.68%). Correlation analyses between microbes and fragrance components showed that *Aspergillus*, *Hyphopichia*, *Trichosporon*, *Alternaria* were all highly and positively correlated with pyrazines, but the dominant fungal communities were highly correlated with only a few individual acid compounds but not with acid compounds overall. The unique Daqu -making process and environment lead to these differences.

## Introduction

Liquors(Baijiu), as treasures of the Chinese nation, are characterized by extraordinary fermentation characteristics, diverse styles, unique flavors after their long development history [1, 2].

**Funding:** This work was supported by the Key Technologies Research and Development Program of Henan Province of China (202102110130), Major Science and Technology Projects of Henan Province of China (181100211400), the Scientific Research Foundation for Docotors of Henan University of Animal Husbandry and Economy (2018HNUAHEDF011) and the Key Subject Projects of Henan University of Animal Husbandry and Economy(C3060020). The funders had no role in study design, data collection and analysis, decision to publish, or preparation of the manuscript.

**Competing interests:** The authors declare that they have no competing interests.

Taorong-type Baijiu are among the 13 flavors of fragrant Baijiu in China and innovatively integrate Luzhou, Maotai, Fen, and sesame flavors. The Taorong flavor caters to numerous customers because of its fragrant, elegant and exquisite liquor, mellow and plump mouthfeel, and clean and bright aftertaste [3]. Taorong-type Baijiu Daqu is a microecological product enriched with microbial communities, bacteria and a complex Daqu fragrance and is capable of saccharification, fermentation and fragrance formation [4, 5]. High-temperature Daqu (maximum starter temperature controlled above 60˚C) and medium-temperature Daqu (maximum starter temperature controlled at 50–60˚C) varieties exist [6–8]. Daqu is mainly produced from wheat through natural inoculation, raw-material starter production, low-temperature fungal culture, high-temperature conversion, and storeroom storage [9, 10]. Daqu is pivotal for microbe inoculation and creates a fermentation environment for liquor-making [11]. Because of differences in Daqu production processes, Daqu from different production temperatures differs in its microbial community composition, which directly affects the compositions of the fermented mash zymophyte system, enzymatic system, and substance system [12]. Hence, the compositional analysis of microbial communities in medium-temperature Daqu and high-temperature Daqu is urgently needed.

Bacteria, fungi and actinomycetes play important roles in Daqu, and fungal communities are pivotal in producing alcohols, enzymes and fragrances [13]. Fungi consist of molds and saccharomycetes. Molds can secrete saccharifying enzymes, lipases, proteases and hydrolases, decompose macromolecular substances in fermented mashes and produce flavor substances and their precursors in liquors; thus, they are closely related to the formation of all flavor substances in liquors [14]. Moreover, saccharomycetes are a critical group of flora that can produce esters, fragrances and alcohols and thereby provide power for fermentation in liquors [15].

Headspace–solid-phase microextraction–gas chromatography–mass spectrometry (HS-SPME-GC-MS) is a practical technique for characterizing volatile components in complex systems [16, 17]. Compared with conventional flavor substance extraction, SPME is more convenient, faster, economically safer, solventless, highly selective, and more sensitive [18]. SPME can be directly combined with GC-MS, high-performance liquid chromatography, or capillary electrophoresis, which largely accelerates analysis and detection [15]. Compared with the traditional culture isolation method, high-throughput sequencing (HTS) can be used for both culturable and unculturable microbes [19]. It has the outstanding characteristics of accurate quantification, high sensitivity, a low workload, and low costs in comparison with fluorescence in situ hybridization, terminal restriction fragment length polymorphism analysis, and 18S rDNA cloning library analysis [8, 19, 20]. Hence, HTS can better determine the compositions of flora communities in samples.

Currently, there has been little research on fungal community compositions and fragrance compositions in medium- and high-temperature Daqu. Herein, HTS was combined with HS-SPME-GC-MS to study Yangshao Taorong-type Baijiu medium-temperature Daqu and high-temperature Daqu. Specifically, the main fungal compositions and fragrance compositions of Taorong-type Baijiu medium-temperature Daqu and high-temperature Daqu were analyzed. The findings will theoretically underlie the fungal flora library establishment and quality identification of Taorong-type Baijiu Daqu.

## Materials and methods

### Materials and reagents

The Daqu used herein was Taorong-type Baijiu Daqu (Henan Yangshao Liquor Co. Ltd.) that had been stored for 5 months. The medium-temperature Daqu (The incubation temperature

is between 50˚C and 60˚C, and the maximum does not exceed 60˚C) and high-temperature Daqu (The incubation temperature is above 60˚C, and the maximum temperature can reach 70˚C) were marked D-Z and E-G, respectively. Daqu was crushed, immediately placed in ice-boxes, transported to the laboratory and stored in a refrigerator at -20˚C. The reagents used herein included anhydrous ethanol (Guangzhou Chemical Reagent Factory), Goldview agarose (Beijing Mengyimei Commercial Center); PCR reagents (Biolion Technology Co. Ltd.); and soil DNA extraction kits (Guangzhou Magen Technology Co. Ltd.). In addition, AMPure XP magnetic beads were purchased from Beckman Instruments (USA).

## Instruments

The instruments used included a -80˚C refrigerator (Zhongke Meiling Cryogenics Co. Ltd.), a 4˚C centrifuge and pipettors (Eppendorf AG, Germany), an ultrapure water apparatus (Shanghai RephiLe Bioscience), a gel-imaging system (Shanghai Biotanon Co. Ltd.), a PCR apparatus (Dongsheng Xingye Science Equipment Co. Ltd.), an agarose gel electrophoresis analyzer (Beijing Liuyi Biotechnology Co. Ltd.), a vortex oscillator (Guangzhou Wego Instrument Co. Ltd.), a GC–MS meter (Shimadzu, Japan), and a solid-phase microextraction device (Merck, USA).

## Experimental methods

**Detection and identification of flavor components.** *Pretreatment of Daqu samples*. At each time point, Daqu (1 g) was placed into a headspace bottle, and 2 g of NaCl and 5 mL of distilled water were added. Then, the bottle was plugged and shaken.

*HS-SPME conditions*. Headspace bottles containing pit mud samples were placed into a water bath at 50˚C for 10 min of preheating, and then solid-phase CAR/PDMS (75 μm CAR/PDMS, carbon molecular sieve/polydimethyl silane) extraction fiber heads were inserted into the silica gel plugs for 30 min of headspace adsorption.

*GC conditions*. An HP-FFAP column ($30 \times 0.32$ mm$^2 \times 0.25$ μm) was used without diffusion at a flow rate of 1.21 mL/min, an inlet temperature of 250˚C, and a temperature increase to 40˚C over 3 min, followed by an increase at 5˚C/min to 60˚C, no heating, and then a temperature increase at 8˚C/min to 230˚C and holding for 7 min.

*MS conditions*. An interface temperature of 220˚C, electron ionization source, electron energy of 70 eV, and ion source temperature of 200˚C were used [21].

**Molecular identification of microorganism communities of Taorong-type Baijiu Daqu by PCR system.** Crushed samples of medium-temperature Daqu or high-temperature Daqu were mixed well (each sample measured 0.25–0.5 g), and total metagenomics were extracted from the microbes using HiPure soil DNA kits. Then, the total DNA quantity and integrity were detected by agarose electrophoresis, and the DNA was stored at -20˚C.

ITS1_plant was amplified by PCR with the primers ITS1_F_KYO2 (5′–TAGAGGAAG–TAAAAGTCGTAA-3′) and ITS86R (5′–TTCAAAGATTCGATGATTCAC-3′).

The first-round amplification procedure was as follows: 1.5 μL of primer R (10 μM), 1.5 μL of primer F (10 μM), 3 μL of 25 mM MgSO$_4$, 1 μL of KOD enzyme, 5 μL of 2 mM dNTPs, 5 μL of 10×Buffer KOD, and X μL of the template (100 ng) were mixed, and the volume was brought up to 50 μL with H$_2$O. The PCR procedure was 94˚C for 2 min; 98˚C for 10 s, 62–66˚C for 30 s, and 68˚C for 30 s (30 cycles); and 68˚C for 5 min. The PCR products were purified using AMPure XP beads. The second-round amplification procedure was as follows: 1 μL of universal PCR primer (10 μM), 1 μL of index primer (10 μM), 3 μL of 25 mM MgSO$_4$, 1 μL of KOD enzyme, 5 μL of 2 mM dNTPs, 5 μL of 10×Buffer KOD, and 1 μL of template (100 ng) were mixed, and the volume was brought up to 50 μL with H$_2$O [7]. The PCR procedure was

as follows: 94˚C for 2 min; 98˚C for 10 s, 65˚C for 30 s, and 68˚C for 30 s (12 cycles); and 68˚C for 5 min.

**Data processing.** After raw reads were acquired from sequencing, the low-quality reads were filtered using FASTP, and double-end reads were spliced into tags using FLASH. Then, the tags were filtered to form clean tags. The clean tags were clustered, and chimeric tags identified during this process were removed using UCHIME from USEARCH, which left the effective tags remaining. Then, the clean tags were clustered, and chimeric tags identified during this process were removed using UCHIME from USEARCH, leaving the remaining effective tags. The abundance of OTUs was statistically analyzed on the based on the effective tags.

## Results

### Rationality analysis of sequencing data

The average numbers of series in the original data of medium-temperature Daqu and high-temperature Daqu were 133251 and 131595, respectively (Table 1). After quality control and filtering, the average numbers of high-quality series were 133170 and 131498, respectively. After the removing of chimeras, the average numbers of effective series were 125899 and 123639, respectively, and the average proportions of effective series were 94.48% and 93.95%, respectively.

### Venn diagram analysis

The numbers of OTUs in medium-temperature Daqu and high-temperature Daqu were 139 and 213, respectively, with 84 shared OTUs. Specifically, medium-temperature Daqu and high-temperature Daqu shared 84 fungal species and possessed 55 and 129 exclusive OTUs (Fig 1).

### Alpha diversity analysis

The average Shannon index of the high-temperature Daqu was larger than that of the medium-temperature Daqu (2.21 vs. 2.01, Table 2), suggesting that the microbial species diversity in the samples of high-temperature Daqu was richer and significantly more complex. The average Chao1 and ACE values of the high-temperature Daqu were both larger than those of the medium-temperature Daqu (268.3 vs. 208.7 and 274.3 vs. 212.1), indicating that the microbial diversity in high-temperature Daqu was higher. In addition, the sample coverage rates of both types of Daqu were larger than 99.9%, indicating that the sequencing results were highly reliable and sufficiently reflected the real situations of the samples.

### Fungal community composition of Daqu

Based on the species annotations of OTUs, the fungal community compositions of each sample at the phylum and genus taxonomic levels were analyzed (Fig 2A and 2B).

**Table 1. Basic sequencing data of medium-temperature Daqu and high-temperature Daqu.**

| Sample | Raw PE | Clean PE | Effective Tags | Effective Ratio |
|--------|--------|----------|----------------|-----------------|
| D-Z-1 | 135860 | 135780 | 128618 | 94.67% |
| D-Z-2 | 131580 | 131502 | 123692 | 94.01% |
| D-Z-3 | 132312 | 132228 | 125387 | 94.77% |
| E-G-1 | 131114 | 131015 | 123134 | 93.91% |
| E-G-2 | 129348 | 129258 | 121116 | 93.64% |
| E-G-3 | 134324 | 134220 | 126666 | 94.30% |

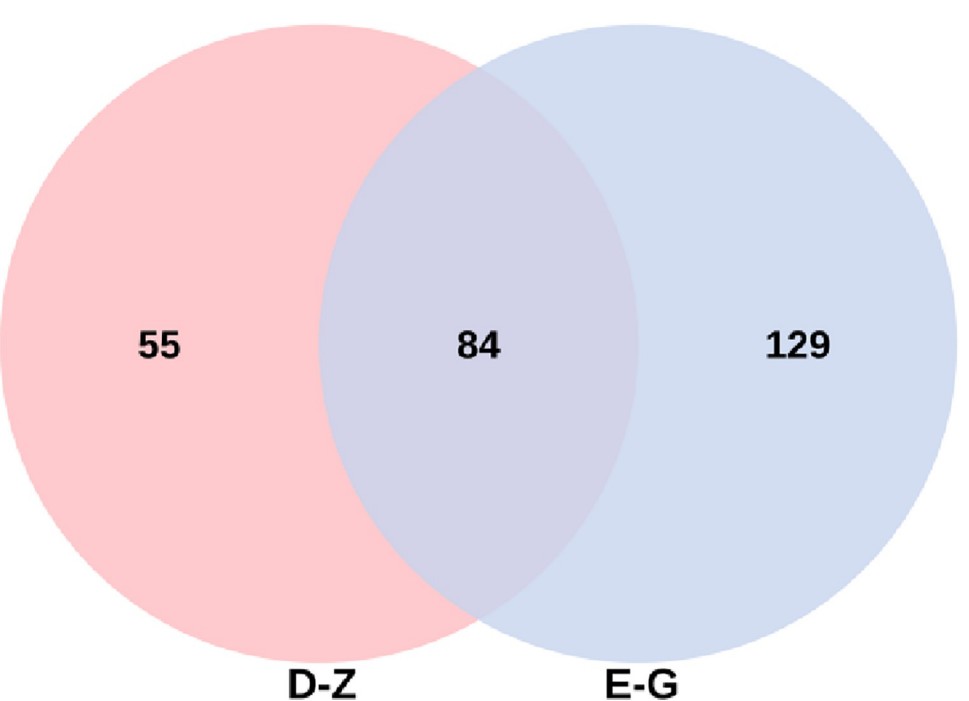

**Fig 1. Venn diagrams of OTUs from medium-temperature Daqu and high-temperature Daqu.**

At the phylum level, with an abundance ≥0.01% as the threshold, 3 phyla, Mucoromycota (72.41%), Ascomycota (27.41%) and Basidiomycota (0.18%), were identified in medium-temperature Daqu (Fig 2(A)). The three phyla Ascomycota (75.51%), Mucoromycota (24.44%) and Basidiomycota (0.04%) were identified in high-temperature Daqu.

At the genus level with an abundance ≥0.01% as the threshold, 15 and 13 genera were recognized in medium- and high-temperature Daqu, respectively. The genera in medium-temperature Daqu included *Rhizopus* (72.40%), *Aspergillus* (13.65%), *Hyphopichia* (3.97%), *Thermoascus* (3.49%), *Thermomyces* (1.30%), *Alternaria* (0.27%), *Trichosporon* (0.15%), *Wickerhamomyces* (0.07%), *Monascus* (0.07%), *Issatchenkia* (0.02%), *Cutaneotrichosporon* (0.01%), *Geosmithia* (0.01%), *Mucor* (0.01%) and *Fusarium* (0.01%). The genera in high-temperature Daqu included *Thermomyces* (53.32%), *Rhizopus* (24.44%), Unclassified (11.91%), *Thermoascus* (7.14%), *Aspergillus* (2.52%), *Monascus* (0.31%), *Hyphopichia* (0.09%), *Alternaria* (0.05%), *Wickerhamomyces* (0.04%), *Cladosporium* (0.03%), *Cyphellophora* (0.02%), *Pseudeurotium* (0.01%) and *Papiliotrema* (0.01%) (Fig 2(B)).

The proportions of *Thermomyces*, *Thermoascus* and *Monascus* in high-temperature Daqu were all larger than those in medium-temperature Daqu, indicating that these three microbial

**Table 2. Medium-temperature Daqu and high-temperature Daqu fungal diversity index table.**

| Sample | Shannon | Simpson | Goods-Coverage | Chao1 | ACE |
|---|---|---|---|---|---|
| D-Z-1 | 2.02 | 0.58 | 99.97% | 194.1 | 205.8 |
| D-Z-2 | 1.99 | 0.57 | 99.97% | 190.7 | 195.6 |
| D-Z-3 | 2.01 | 0.58 | 99.99% | 241.1 | 234.9 |
| E-G-1 | 2.21 | 0.67 | 99.95% | 266.4 | 270.8 |
| E-G-2 | 2.17 | 0.66 | 99.95% | 254.2 | 263.0 |
| E-G-3 | 2.27 | 0.66 | 99.95% | 284.2 | 289.0 |

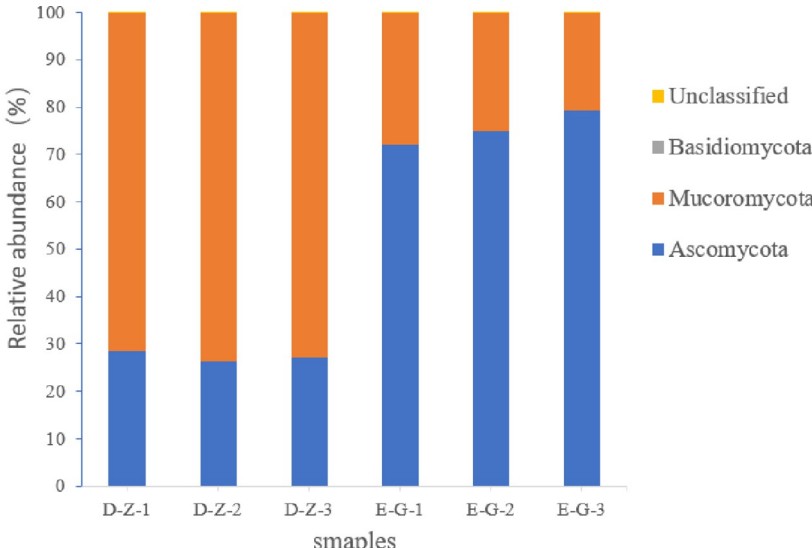

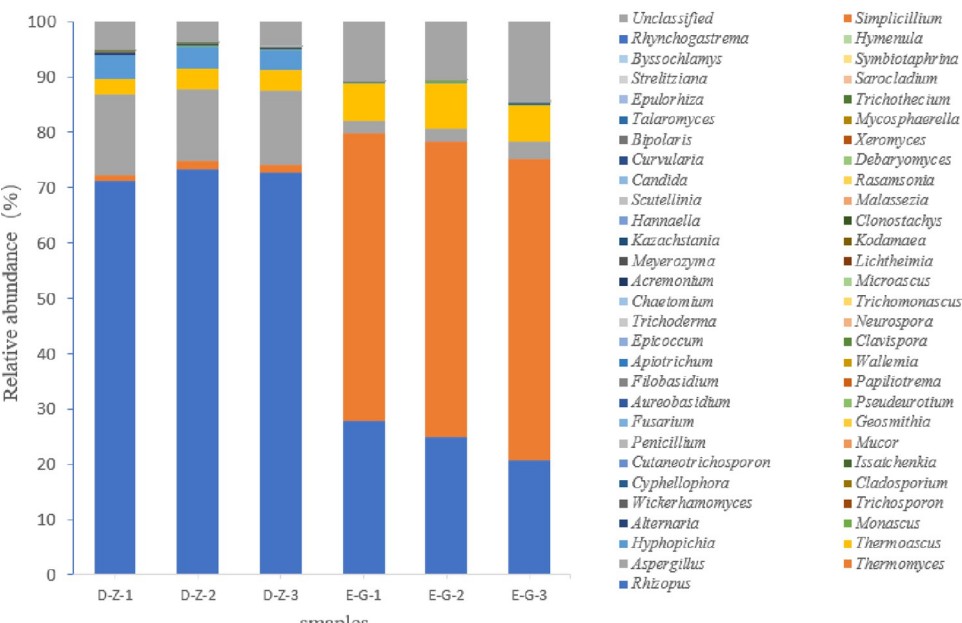

**Fig 2.** (A) Fungal taxonomy of medium- and high-temperature Daqu at the phylum level. (B) Fungal taxonomy of medium- and high-temperature Daqu at the genus level.

genera are highly heat-resistant and adapted to hot conditions. The proportions of *Rhizopus*, *Aspergillus*, *Hyphopichia*, *Alternaria*, *Trichosporon*, *Wickerhamomyces* and *Mucor* all decreased with increasing starter temperatures, indicating that these microbial genera are adapted to medium-temperature environments. In addition, three genera, *Cyphellophora*, *Pseudeurotium* and *Papiliotrema*, were only identified in high-temperature Daqu, suggesting that these three genera are thermally resistant and can survive hot conditions. *Thermomyces* can thrive at up to 60˚C but cannot survive below 20˚C [22, 23]. They can secrete thermophilic enzymes, improve

the catalytic efficiency of reactions owing to their high thermal stability, and facilitate binding between enzymes and substrates owing to their low viscosity in substrates under hot conditions, which are all advantages for catalysis by thermophilic mannanase [13]. *Thermoascus* can produce heat- and acid-resistant xylanase, which adapts to 70˚C and pH 4.8 [24]. *Monascus* can produce various enzymes during the growth process, and their high enzymatic esterifying activity can catalyze the synthesis of acids and alcohols that are significant sources of fragrance in liquors [12]. Moreover, *Rhizopus* can generate abundant highly active amylases, proteases and lipases and can produce citric acid, gluconic acid, lactic acid, succinic acid and other organic acids [25]. *Aspergillus*, which is pivotal in liquor-making, can secrete saccharifying enzymes and amylases and facilitate the fermentation of Daqu, which is favorable for liquor production by fermentation [6]. Xue et al. [26] studied the microbial diversity in Daqu from Xiangyang Shihua liquors using HTS. The dominant fungal phyla in the Daqu (abundance >0.1%) were Ascomycota (99.70%), Basidiomycota (0.17%) and Mucoromycota (0.12%), and the dominant fungal genera (> 1%) included *Saccharomycopsis* (57.08%), *Monascus* (17.58%), *Thermoascus* (5.38%), *Penicillium* (4.76%), *Aspergillus* (1.78%) and *Leiothecium* (1.22%). Their findings are consistent with our results regarding phyla and genera. Zhou et al. [27] investigated the microbial community compositions in medium–high temperature Daqu and the Daqu-making environment by HTS and detected 31 genera in the Daqu and after fermentation. In particular, *Aspergillus* and *Rhizopus* were the most abundant in the Daqu, and upon fermentation, the dominant genera in the Daqu included *Aspergillus* (39.14±0.21%), *Candida* (6.39±0.31%) and *Rhizopus* (0.3739±0.01%). After fermentation, *Thermoascus* gradually became another dominant genus (23.96±0.01%). Hence, the fungal genera found in Daqu and during fermentation are consistent with our results. Chen et al. [28] analyzed the microbial community compositions in Taorong-type Baijiu Daqu by HTS and found that except for the unclassified genera in 4 Daqu samples, other dominant fungal genera were *Thermoascus* (44.07%), *Aspergillus* (10.04%), *Thermomyces* (3.79%), *Alternaria* (1.78%), *Emericella* (1.02%), *Wickerhamomyces* (0.78%) and *Rhizomucor* (0.42%). Their findings on fungal genera are consistent with our results. The results of the above studies are similar to our HTS results. Nevertheless, we identified more fungal communities and found *Pseudeurotium* and *Papiliotrema* for the first time in the medium-temperature Daqu and high-temperature Daqu of Taorong-type Baijiu starters.

## Fragrance component analysis of medium-temperature and high-temperature Daqu

In total, 40 and 29 fragrance components were identified in medium- temperature and high-temperature Yangshao Daqu, respectively (Table 3). Medium-temperature Daqu contained 14 acids, 8 esters, 4 pyrazines, 4 alcohols, 3 phenols, 2 aromatics, 2 aldehydes, 1 alkene, 1 ketone and 1 aromatic hydrocarbon, with average relative concentrations of 27.1%, 2.11%, 53.12%, 5.42%, 1.04%, 1.99%, 0.52%, 2%, 0.47% and 0.66%, respectively. High-temperature Daqu contained 9 acids, 5 alcohols, 4 esters, 2 pyrazines, 2 aromatics, 2 phenols, 2 aldehydes, 1 alkene, 1 ketone, and 1 aromatic hydrocarbon, with average relative concentrations of 32.68%, 8.12%, 3.07%, 24.4%, 3.17%, 2.91%, 1.81%, 4.06%, 1.92% and 1.38%, respectively.

These results indicate that alcohols, pyrazines, esters, acids and phenols are diverse and abundant in Daqu. Alcohols are a major class of flavor substances in liquors and are the precursors of esters. Alcohols can highlight the fragrances of esters and make liquors taste mellow, which together contribute to long-lasting fragrances [29]. Pyrazines are capable of expanding blood vessels, improving blood circulation and protecting the liver (preventing alcohols from injuring the gastric mucosa and liver) [30]. Esters mostly have aromatic smells that endow

**Table 3. Fragrance components.**

| Type | No. | Medium-temperature Daqu (D-Z) Compound name | Relative percentage (%) | | | High-temperature Daqu (E-G) Compound name | Relative percentage (%) | | |
|---|---|---|---|---|---|---|---|---|---|
| Alcohols | R-A | phenylethanol | 3.47 | 3.51 | 3.52 | phenylethanol | 3.80 | 3.81 | 3.82 |
| | R-B | cineole | 0.69 | 0.71 | 0.72 | cineole | 1.70 | 1.73 | 1.73 |
| | R-C | benzyl alcohol | 0.53 | 0.54 | 0.55 | benzyl alcohol | 0.85 | 0.84 | 0.86 |
| | R-D | 3,6,9,12-tetradecane-1-methanol | 0.66 | 0.69 | 0.66 | — | — | — | — |
| | R-E | — | — | — | — | tetrahydro-2,5-dimethyl-2hydro-pyran methanol | 1.06 | 1.08 | 1.07 |
| | R-F | — | — | — | — | 4-methylene-6-hepten-2-methanol | 0.66 | 0.69 | 0.69 |
| Pyrazines | S-A | tetramethylpyrazine | 46.88 | 46.83 | 46.84 | tetramethylpyrazine | 22.60 | 22.59 | 22.58 |
| | S-B | trimethylpyrazine | 5.40 | 5.38 | 5.39 | trimethylpyrazine | 1.80 | 1.80 | 1.82 |
| | S-C | 2,3-dimethylpyrazine | 0.52 | 0.49 | 0.49 | — | — | — | — |
| | S-D | 2,3,5- trimethyl-6-ethylpyrazine | 0.42 | 0.38 | 0.37 | — | — | — | — |
| Acids | T-A | — | — | — | — | 3-methyl butyric acid | 13.30 | 13.28 | 13.29 |
| | T-B | valeric acid | 11.78 | 11.80 | 11.79 | valeric acid | 0.25 | 0.26 | 0.24 |
| | T-C | hexanoic acid | 5.29 | 5.31 | 5.27 | hexanoic acid | 12.47 | 12.45 | 12.46 |
| | T-D | acetic acid | 2.65 | 2.63 | 2.67 | acetic acid | 2.39 | 2.37 | 2.35 |
| | T-E | oleic acid | 2.60 | 2.56 | 2.55 | — | — | — | — |
| | T-F | stearic acid | 1.44 | 1.44 | 1.47 | — | — | — | — |
| | T-G | butyric acid | 1.10 | 1.06 | 1.05 | butyric acid | 1.07 | 1.08 | 1.09 |
| | T-H | n-palmitic acid | 0.68 | 0.65 | 0.65 | n-palmitic acid | 0.84 | 0.82 | 0.83 |
| | T-I | octanoic acid | 0.45 | 0.42 | 0.42 | octanoic acid | 1.04 | 1.01 | 1.01 |
| | T-J | 4-methylvaleric acid | 0.36 | 0.33 | 0.36 | 4-methylvaleric acid | 0.90 | 0.92 | 0.91 |
| | T-K | heptanoic acid | 0.24 | 0.22 | 0.23 | heptanoic acid | 0.46 | 0.49 | 0.46 |
| | T-L | (R)-(-)-4-methylhexanoic acid | 0.21 | 0.23 | 0.22 | — | — | — | — |
| | T-M | nonanoic acid | 0.18 | 0.19 | 0.17 | — | — | — | — |
| | T-N | 3-methyl-2-crotonic acid | 0.17 | 0.16 | 0.12 | — | — | — | — |
| | T-O | 2-methyl-1-methyl propyl-butyric acid | 0.06 | 0.08 | 0.10 | — | — | — | — |
| Esters | U-A | 2-isobutoxy ethyl butyrate | 0.51 | 0.48 | 0.48 | — | — | — | — |
| | U-B | 2,4-dimethyl-3-isobutyl carbonic ester | 0.34 | 0.37 | 0.37 | — | — | — | — |
| | U-C | ethyl phenylacetate | 0.30 | 0.27 | 0.30 | — | — | — | — |
| | U-D | 2-valeric-ethoxyethyl | 0.27 | 0.28 | 0.29 | — | — | — | — |
| | U-E | ethyl 3-phenylpropionate | 0.27 | 0.25 | 0.26 | ethyl 3-phenylpropionate | 0.46 | 0.43 | 0.46 |
| | U-F | 1-methoxyl-2-propyl acetate | 0.24 | 0.24 | 0.21 | — | — | — | — |
| | U-G | 2-ethoxy ethyl 2-methyl butyrate | 0.13 | 0.10 | 0.13 | — | — | — | — |
| | U-H | butyric-2-methyl-1-methyl propyl | 0.07 | 0.09 | 0.08 | — | — | — | — |
| | U-I | — | — | — | — | ethyl caproate | 1.81 | 1.82 | 1.80 |
| | U-J | — | — | — | — | ethyl palmitic | 0.43 | 0.42 | 0.41 |
| | U-K | — | — | — | — | ethyl phenylacetate | 0.37 | 0.39 | 0.41 |
| Phenols | V-A | 2,4-bi(1,1-dimethyl ethyl)-phenol | 0.62 | 0.67 | 0.66 | 2,4-bi(1,1-dimethyl ethyl)-phenol | 1.76 | 1.75 | 1.74 |
| | V-B | 2-methoxyl-4-vinyl phenol | 0.22 | 0.19 | 0.19 | — | — | — | — |
| | V-C | p-cresol | 0.17 | 0.19 | 0.21 | p-cresol | 1.16 | 1.17 | 1.18 |
| Aromatics | W-A | p-xylene | 1.71 | 1.72 | 1.73 | p-xylene | 2.65 | 2.66 | 2.64 |
| | W-B | 1,4-diethylbenzene | 0.26 | 0.27 | 0.28 | 1,4-diethylbenzene | 0.50 | 0.53 | 0.50 |
| Aldehydes | X-A | 1H-pyrrole-2-formaldehyde | 0.34 | 0.36 | 0.35 | 1H-pyrrole-2-formaldehyde | 0.87 | 0.88 | 0.86 |
| | X-B | benzaldehyde | 0.19 | 0.15 | 0.17 | benzaldehyde | 0.96 | 0.94 | 0.95 |
| Alkenes | Y-A | D-limonene | 2.0 | 1.9 | 2.1 | D-limonene | 4.05 | 4.08 | 4.05 |
| Ketones | Z-A | 1-(1H-pyrrole-2-acetyl)-ethyl ketone | 0.46 | 0.46 | 0.49 | 1-(1H-pyrrole-2-acetyl)-ethyl ketone | 1.93 | 1.93 | 1.90 |
| Aromatic hydrocarbons | A-A | o-isopropyl phenylmethane | 0.65 | 0.67 | 0.66 | o-isopropyl phenylmethane | 1.37 | 1.38 | 1.39 |

liquors with fruit fragrances and are the key substances that determine the styles and quality of liquors [22]. Acids at appropriate concentrations can increase the brightness of liquors and remove dry and spicy tastes, making liquors smoother [8]. Phenols endow liquors with unique sooty, burnt, milky and pit mud tastes and are major components of the dominant flavors of liquors [31].

### Heatmap of the species distribution at the genus level and correlation analysis of microbes and fragrance compositions

A redder color indicates a higher proportion of a given species in Fig 3(A). In medium-temperature Daqu, the relatively abundant genera included *Rhizopus*, *Aspergillus*, *Hyphopichia*, *Thermoascus* and *Thermomyces*. In high-temperature Daqu, the relatively abundant genera were *Thermomyces*, *Thermoascus* and *Monascus*.

The Spearman correlation coefficients of microbes and flavor components were calculated, and thereby, the correlations of the qualified data were plotted. *Rhizopus* was largely and positively correlated with 3,6,9,12-tetradecane-1-ethanol, valeric acid, and (R)-(-)-4-methylhexanoic acid (Fig 3(B)). *Aspergillus* was positively correlated with pyrazines, 2-isobutoxy ethyl butyrate, and ethyl phenylacetate, showing the largest correlation with trimethyl pyrazine. *Hyphopichia* was positively correlated with pyrazines, showing the largest correlations with 2,3-dimethyl pyrazine and 2,3,5-trimethyl-6-ethyl pyrazine, which are two compounds that were only found in medium-temperature Daqu. *Thermoascus* was negatively but not highly correlated with pyrazines and positively correlated with acids and was most positively correlated with 2-ethoxy ethyl-2-methyl butyrate. *Thermomyces* was negatively correlated with pyrazines, and except for a negative correlation with 3,6,9,12-tetradecane-1-ethanol, it was highly and positively correlated with other types of acids, although the correlation with overall acids was low. *Alternaria* was positively correlated with pyrazines and most highly and positively correlated with 3,6,9,12-tetradecane-1-ethanol and 1-methoxyl-2-propyl acetate. *Trichosporon* was positively correlated with pyrazines and very positively correlated with 3,6,9,12-tetradecane-1-ethanol, oleic acid, 2-isobutoxy ethyl butyrate, and 1-methoxyl-2-propyl acetate.

### Conclusion

The fungal community compositions and fragrance components in Yangshao medium-temperature Daqu and high-temperature Daqu were studied. At the threshold of ≥0.01% abundance, three phyla were identified in both medium-temperature Daqu and high-temperature Daqu: Mucoromycota, Ascomycota and Basidiomycota. At the genus level, 15 and 13 genera were recognized in medium- and high-temperature Daqu, respectively. *Rhizopus* (72.40%) and *Thermomyces* (53.32%) accounted for the largest proportions in medium-temperature Daqu and high-temperature Daqu, respectively. In addition, *Cyphellophora*, *Pseudeurotium* and *Papiliotrema* were identified in high-temperature Daqu but were not detected in medium-temperature Daqu. GC identified 40 and 29 fragrance components in Yangshao medium-temperature Daqu and high-temperature Daqu, respectively, which contained the highest proportions of pyrazines (53.12%) and acids (32.68%), respectively. Correlation analysis of microbes and fragrance compositions showed that pyrazines were highly correlated with *Aspergillus*, *Trichosporon*, *Hyphopichia* and *Alternaria*. Acids were generally not highly correlated with the dominant flora, but high correlations were found for oleic acid with *Trichosporon* and *Hyphopichia* and for valeric acid and (R)-(-)-4-methylhexanoic acid with *Rhizopus*.

The unique starter production process and flora environment are the fundamental causes of the diversity of fungal flora in Yangshao Taorong-type Baijiu Daqu, and fragrance components are an important standard for the evaluation of Daqu and finished liquor products.

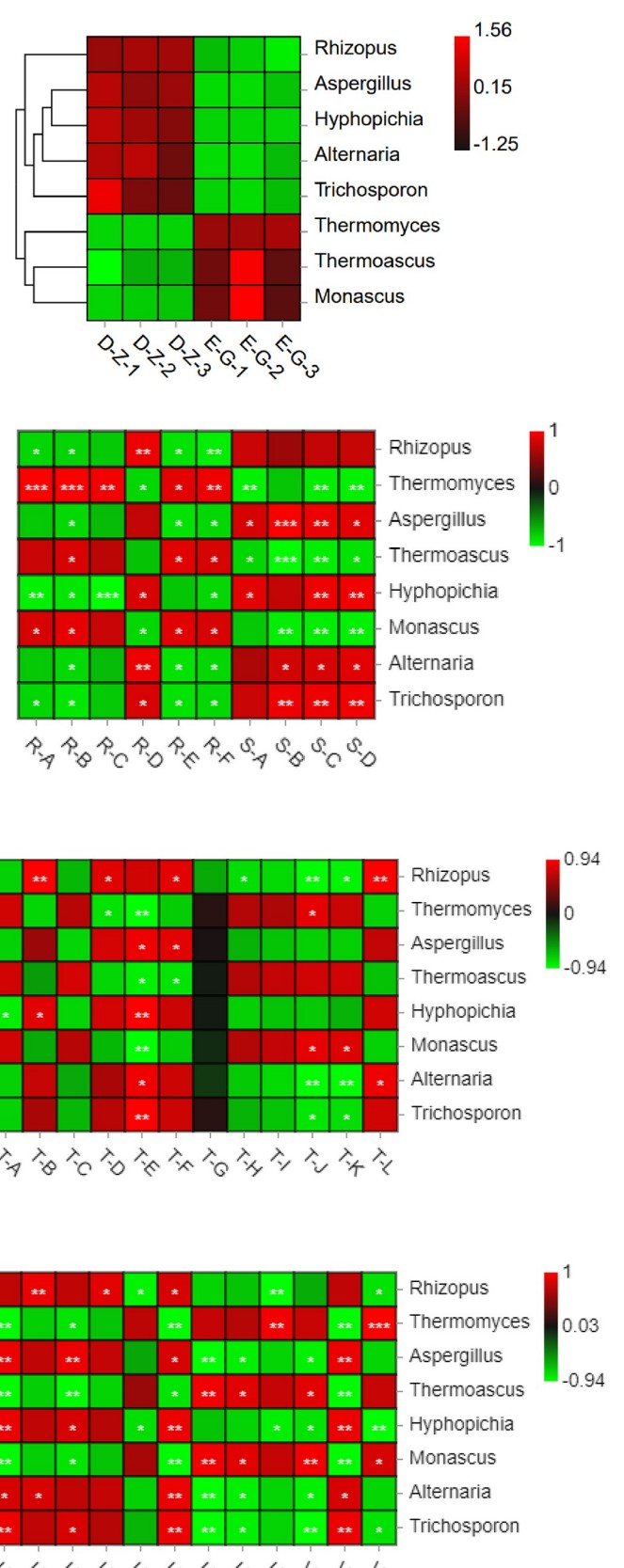

**Fig 3.** (A) Heatmap of the species distribution at the genus level. (B) Calculated Spearman correlation coefficients of microbes and flavor components and the corresponding plotted correlations of the qualified data.

Thus, HTS and HS–SPME–GC–MS were combined to analyze the major fungal compositions and fragrance components in medium-temperature and high-temperature Taorong-type Baijiu Daqu. The findings of this study will theoretically underlie the fungal flora library establishment and quality identification of Taorong-type Baijiu Daqu.

## Acknowledgments

We would like to thank American Journal Experts for English language editing.

## Author Contributions

**Conceptualization:** Yanbo Liu.

**Project administration:** Chunmei Pan.

**Writing – original draft:** Xin Li.

**Writing – review & editing:** Yanbo Liu, Haideng Li, Huimin Zhang, Xiangkun Shen, Lixin Zhang, Suna Han, Chunmei Pan.

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
