## [Decision Letter · Decision Letter 0]

18 Jul 2022

PONE-D-22-11830Fungal Community Compositional Diversity and Fragrance Components in Medium- and High-Temperature          Taorong-Type DaquPLOS ONE

Dear Dr. Liu,

Thank you for submitting your manuscript to PLOS ONE. After careful consideration, we feel that it has merit but does not fully meet PLOS ONE’s publication criteria as it currently stands. Therefore, we invite you to submit a revised version of the manuscript that addresses the points raised during the review process.

Please, It is requested that all comments and suggestions from reviewers be addressed and review in detail the authors' guide. Please submit your revised manuscript by Sep 01 2022 11:59PM. If you will need more time than this to complete your revisions, please reply to this message or contact the journal office at plosone@plos.org. Please include the following items when submitting your revised manuscript:A rebuttal letter that responds to each point raised by the academic editor and reviewer(s). You should upload this letter as a separate file labeled 'Response to Reviewers'.A marked-up copy of your manuscript that highlights changes made to the original version. You should upload this as a separate file labeled 'Revised Manuscript with Track Changes'.An unmarked version of your revised paper without tracked changes. You should upload this as a separate file labeled 'Manuscript'.If applicable, we recommend that you deposit your laboratory protocols in protocols.io to enhance the reproducibility of your results. Protocols.io assigns your protocol its own identifier (DOI) so that it can be cited independently in the future. For instructions see: https://journals.plos.org/plosone/s/submission-guidelines#loc-laboratory-protocols. Additionally, PLOS ONE offers an option for publishing peer-reviewed Lab Protocol articles, which describe protocols hosted on protocols.io. Read more information on sharing protocols at https://plos.org/protocols?utm_medium=editorial-email&utm_source=authorletters&utm_campaign=protocols.

We look forward to receiving your revised manuscript.

Kind regards,

Mónica L. Chávez-González, PhD

Academic Editor

PLOS ONE

Journal Requirements:

 “Funding

This work was supported by the Key Technologies Research and Development Program of Henan Province of China (202102110130), Major Science and Technology Projects of Henan Province of China (181100211400), the Scientific Research Foundation for Docotors of Henan University of Animal Husbandry and Economy (2018HNUAHEDF011) and the Key Subject Projects of Henan University of Animal Husbandry and Economy(C3060020).”

“This work was supported by the Key Technologies Research and Development Program of Henan Province of China (202102110130), Major Science and Technology Projects of Henan Province of China (181100211400), the Scientific Research Foundation for Docotors of Henan University of Animal Husbandry and Economy (2018HNUAHEDF011) and the Key Subject Projects of Henan University of Animal Husbandry and Economy(C3060020).”

 “Funding

This work was supported by the Key Technologies Research and Development Program of Henan Province of China (202102110130), Major Science and Technology Projects of Henan Province of China (181100211400), the Scientific Research Foundation for Docotors of Henan University of Animal Husbandry and Economy (2018HNUAHEDF011) and the Key Subject Projects of Henan University of Animal Husbandry and Economy(C3060020). “

6. Please ensure that you refer to Figure 1 in your text as, if accepted, production will need this reference to link the reader to the figure.

Additional Editor Comments:

It is requested that all comments and suggestions from reviewers be addressed.

Reviewers' comments:

Reviewer's Responses to Questions

**Comments to the Author**

1. Is the manuscript technically sound, and do the data support the conclusions?

Reviewer #1: Yes

Reviewer #2: Yes

2. Has the statistical analysis been performed appropriately and rigorously? 

Reviewer #1: Yes

Reviewer #2: Yes

3. Have the authors made all data underlying the findings in their manuscript fully available?

Reviewer #1: Yes

Reviewer #2: Yes

4. Is the manuscript presented in an intelligible fashion and written in standard English?

Reviewer #1: Yes

Reviewer #2: Yes

5. Review Comments to the Author

Reviewer #1: The analyzed document meets the requirements to be published as an innovation article in its area and demonstrates having an impact on the development of new research hypotheses. However, it is recommended to publish reviewing the grammatical review considerations because although the review is appreciated, the writing has spelling and writing errors and some other grammatical signs.

Similar investigations were searched, and they meet the profile.

observations

Adscriptions

Checking for double spaces and spaces after commas is required

Abstract

Line 24: In the results section The word largest is often overused in the text, its recommended the use of the most significant.

Line 26: respectively, It appears that you have an unnecessary comma in a compound predicate. Consider removing it.

Line 29: , and. Your sentence contains a series of three or more words, phrases, or clauses. Consider inserting a comma to separate the elements.

Keywords

Line 35: , Fungal It appears that you have improperly spaced some punctuation. Consider adding a space.

Introduction

Well done, only consider:

Line 37: , and. Your sentence contains a series of three or more words, phrases, or clauses. Consider inserting a comma to separate the elements.

Line 53: fpr change by for

Line 64: It seems that you have an unnecessary comma. Consider removing the comma.

Materials and Methods

Line 95: iceboxes The word ice boxes seems to be miswritten. Consider replacing it.

Line 97: It seems that semicolon use may be incorrect here. Consider using a comma instead of a semicolon

Line 99: Consider adding a transition phrase to improve the flow of your paragraph. In addition, AMPure XP…

Line 140: It appears that then may be unnecessary in this sentence. Consider removing it.

Line 143: The noun phrase template seems to be missing a determiner before it. Consider adding an article.

Line 149: Your sentence may be unclear or hard to follow. Consider rephrasing. (Option: The clean tags were clustered, and chimeric tags identified during this process were removed using UCHIME from USEARCH, leaving the remaining effective tags)

Line 153: The phrase on the basis of may be wordy. Consider changing the wording (based on)

Results

Line 161: The phrase the removal of may be wordy. Consider changing the wording. (removing)

Line 171: Delete respectively.

Line 222: Change: Monascus can produce various enzymes during the growth process, and their high enzymatic esterifying activity can catalyze the synthesis of acids and alcohols that are significant sources of fragrance in liquors

Funding

Line 347: Doctors

Reviewer #2: ABSTRACT

The full title is confusing, the Taorong-Type Daqu, it is liquor flavor and Daqu is a type or fermentation, it must be written in English which is the language of the article and improve writing of the title to link the objective with the liquor. Must be understood in English by all readers. Maybe Fermented liquor flavor Taorong type Daqu or something. The same for the Short title

Clarify in methods molecular identification of microorganism communities and characterization of the volatile components of the fermentation liquor Taorong flavor type Daqu

Keywords, should not be words included in the title

INTRODUCTION

Review writing errors and spaces in text

Unify the format of references in terms of spaces, score, etc.

METHODOLOGY

Clarify that a fermentation liquor that was used, is a micro-ecological product enriched with communities of microorganisms called Daqu Taorong- flavor or something and then name this Taorong-flavored Daqu. First explain, what were the temperatures medium and high

The subtitle Total DNA extraction and quantitative pcr…. it is suggested to change to molecular identification of microorganism communities of taorong-flavored daqu liquor fermentation by PCR system

RESULTS

reference the table in the text

first goes the explanation, discussion of results and then the table

separate discussions from conclusions, discuss in the results

6. PLOS authors have the option to publish the peer review history of their article (what does this mean?). If published, this will include your full peer review and any attached files.

Reviewer #1: **Yes: **Roberto Arredondo-Valdés

Reviewer #2: **Yes: **MIRIAM DESIREE DAVILA MEDINA

---

## [Author Response · Author response to Decision Letter 0]

3 Aug 2022

Dear Editor and Reviewers:

Thank you again for your letter and for the reviewers' comments concerning our manuscript. Those comments are all valuable and very helpful for revising and improving our paper, as well as the important guiding significance to our researches. We have studied comments carefully and have made correction which we hope meet with approval. The main corrections in the paper and the responds to the reviewer 's comments are as following:

Journal Requirements:

It has been modified.

It has been modified.

 “Funding

This work was supported by the Key Technologies Research and Development Program of Henan Province of China (202102110130), Major Science and Technology Projects of Henan Province of China (181100211400), the Scientific Research Foundation for Docotors of Henan University of Animal Husbandry and Economy (2018HNUAHEDF011) and the Key Subject Projects of Henan University of Animal Husbandry and Economy(C3060020).”

It has been modified. I have added after the funding “all awarded to YL. The funders had no role in study design, data collection and analysis, decision to publish, or preparation of the manuscript.”

“This work was supported by the Key Technologies Research and Development Program of Henan Province of China (202102110130), Major Science and Technology Projects of Henan Province of China (181100211400), the Scientific Research Foundation for Docotors of Henan University of Animal Husbandry and Economy (2018HNUAHEDF011) and the Key Subject Projects of Henan University of Animal Husbandry and Economy(C3060020).”

 “Funding

This work was supported by the Key Technologies Research and Development Program of Henan Province of China (202102110130), Major Science and Technology Projects of Henan Province of China (181100211400), the Scientific Research Foundation for Docotors of Henan University of Animal Husbandry and Economy (2018HNUAHEDF011) and the Key Subject Projects of Henan University of Animal Husbandry and Economy(C3060020). “

I have removed Funding in the manuscript. please show it elsewhere.

Relevant data has been uploaded to NCBI database，and the accession numbers is PRJNA861706.

6. Please ensure that you refer to Figure 1 in your text as, if accepted, production will need this reference to link the reader to the figure.

It has been modified.

It has been modified.

Response to Reviewer #1:

Abstract

Line 24: In the results section The word largest is often overused in the text, its recommended the use of the most significant.

It has been modified.

Line 26: respectively, It appears that you have an unnecessary comma in a compound predicate. Consider removing it.

It has been modified.

Line 29: , and. Your sentence contains a series of three or more words, phrases, or clauses. Consider inserting a comma to separate the elements.

It has been modified.

Keywords

Line 35: , Fungal It appears that you have improperly spaced some punctuation. Consider adding a space.

It has been modified.

Introduction

Well done, only consider:

Line 37: , and. Your sentence contains a series of three or more words, phrases, or clauses. Consider inserting a comma to separate the elements.

It has been modified.

Line 53: fpr change by for 

It has been modified.

Line 64: It seems that you have an unnecessary comma. Consider removing the comma.

It has been modified.

Materials and Methods

Line 95: iceboxes The word ice boxes seems to be miswritten. Consider replacing it.

It has been modified.

Line 97: It seems that semicolon use may be incorrect here. Consider using a comma instead of a semicolon

It has been modified.

Line 99: Consider adding a transition phrase to improve the flow of your paragraph. In addition, AMPure XP…

It has been modified.

Line 140: It appears that then may be unnecessary in this sentence. Consider removing it.

It has been modified.

Line 143: The noun phrase template seems to be missing a determiner before it. Consider adding an article.

It has been modified.

Line 149: Your sentence may be unclear or hard to follow. Consider rephrasing. (Option: The clean tags were clustered, and chimeric tags identified during this process were removed using UCHIME from USEARCH, leaving the remaining effective tags)

It has been modified.

Line 153: The phrase on the basis of may be wordy. Consider changing the wording (based on)

Results

It has been modified.

Line 161: The phrase the removal of may be wordy. Consider changing the wording. (removing)

It has been modified.

Line 171: Delete respectively.

It has been modified.

Line 222: Change: Monascus can produce various enzymes during the growth process, and their high enzymatic esterifying activity can catalyze the synthesis of acids and alcohols that are significant sources of fragrance in liquors

It has been modified.

Funding

Line 347: Doctors

It has been modified.

Response to Reviewer #2: 

ABSTRACT

The full title is confusing, the Taorong-Type Daqu, it is liquor flavor and Daqu is a type or fermentation, it must be written in English which is the language of the article and improve writing of the title to link the objective with the liquor. Must be understood in English by all readers. Maybe Fermented liquor flavor Taorong type Daqu or something. 

The same for the Short title

Clarify in methods molecular identification of microorganism communities and characterization of the volatile components of the fermentation liquor Taorong flavor type Daqu

Keywords, should not be words included in the title

Revised and improved the title, new title: Taorong-type Baijiu Starter: Analysis of Fungal Community and Metabolic Characteristics of Middle-Temperature Daqu and High-Temperature Daqu. Redefined keywords: Correlation, Baijiu starter, Flavor ingredient, Microbial community.

INTRODUCTION

Review writing errors and spaces in text

Unify the format of references in terms of spaces, score, etc.

The above notes have been modified.

METHODOLOGY

Clarify that a fermentation liquor that was used, is a micro-ecological product enriched with communities of microorganisms called Daqu Taorong- flavor or something and then name this Taorong-flavored Daqu. First explain, what were the temperatures medium and high

The medium-temperature Daqu (The incubation temperature is between 50°C and 60°C, and the maximum does not exceed 60°C) and high-temperature Daqu (The incubation temperature is above 60°C, and the maximum temperature can reach 70°C) were marked D-Z and E-G, respectively.

The subtitle Total DNA extraction and quantitative pcr…. it is suggested to change to molecular identification of microorganism communities of taorong-flavored daqu liquor fermentation by PCR system

It has been modified.

RESULTS

reference the table in the text

first goes the explanation, discussion of results and then the table

separate discussions from conclusions, discuss in the results

It has been modified.

 We tried our best to improve the manuscript and made some changes in the manuscript. These changes will not influence the content and framework of the paper. We appreciate for Editor and Reviewers' warm work earnestly, and hope that the correction will meet with approval. If there are any shortcomings in the article, please tell me immediately, and I will seriously revise it again.Once again, thank you very much for your comments and suggestion.

Thank you and best regards.

Yours sincerely,

Yanbo Liu E-mail: yanboliu@hnuahe.edu.cn

---

## [Editor Report · Decision Letter 1]

7 Sep 2022

Taorong-type Baijiu Starter: Analysis of Fungal Community and Metabolic Characteristics of Middle-Temperature Daqu and High-Temperature Daqu

PONE-D-22-11830R1

Dear Dr. Yanbo Liu

We’re pleased to inform you that your manuscript has been judged scientifically suitable for publication and will be formally accepted for publication once it meets all outstanding technical requirements.

Kind regards,

Mónica L. Chávez-González, PhD

Academic Editor

PLOS ONE
---

## [Editor Report · Acceptance letter]

26 Sep 2022

PONE-D-22-11830R1 

Taorong-type Baijiu Starter: Analysis of Fungal Community and Metabolic Characteristics of Middle-Temperature Daqu and High-Temperature Daqu 

Dear Dr. Liu:

I'm pleased to inform you that your manuscript has been deemed suitable for publication in PLOS ONE. Congratulations! Your manuscript is now with our production department. 

Kind regards, 

on behalf of

Dr. Mónica L. Chávez-González 

Academic Editor

PLOS ONE